# Development and validation of a prognostic and predictive 32-gene signature for gastric cancer

Jae-Ho Cheong [1,2,3,13✉], Sam C. Wang [4,13], Sunho Park[5,13], Matthew R. Porembka[4], Alana L. Christie[6], Hyunki Kim [7], Hyo Song Kim[8], Hong Zhu [6], Woo Jin Hyung[1], Sung Hoon Noh [1], Bo Hu [9], Changjin Hong[5], John D. Karalis[4], In-Ho Kim[10], Sung Hak Lee[11] & Tae Hyun Hwang [5,12✉]

Genomic profiling can provide prognostic and predictive information to guide clinical care. Biomarkers that reliably predict patient response to chemotherapy and immune checkpoint inhibition in gastric cancer are lacking. In this retrospective analysis, we use our machine learning algorithm NTriPath to identify a gastric-cancer specific 32-gene signature. Using unsupervised clustering on expression levels of these 32 genes in tumors from 567 patients, we identify four molecular subtypes that are prognostic for survival. We then built a support vector machine with linear kernel to generate a risk score that is prognostic for five-year overall survival and validate the risk score using three independent datasets. We also find that the molecular subtypes predict response to adjuvant 5-fluorouracil and platinum therapy after gastrectomy and to immune checkpoint inhibitors in patients with metastatic or recurrent disease. In sum, we show that the 32-gene signature is a promising prognostic and predictive biomarker to guide the clinical care of gastric cancer patients and should be validated using large patient cohorts in a prospective manner.

[1] Department of Surgery, Yonsei University College of Medicine, Seoul, South Korea. [2] Department of Biochemistry and Molecular Biology, Yonsei University College of Medicine, Seoul, South Korea. [3] Department of Biomedical Systems Informatics, Yonsei University College of Medicine, Seoul, South Korea. [4] Division of Surgical Oncology, Department of Surgery, University of Texas Southwestern Medical Center, Dallas, TX, USA. [5] Department of Artificial Intelligence and Informatics, Mayo Clinic, Jacksonville, FL, USA. [6] Department of Clinical Sciences, University of Texas Southwestern Medical Center, Dallas, TX, USA. [7] Department of Pathology, Yonsei University College of Medicine, Seoul, South Korea. [8] Department of Internal Medicine, Division of Medical Oncology, Yonsei University College of Medicine, Seoul, South Korea. [9] Department of Quantitative Health Sciences, Lerner Research Institute, Cleveland Clinic, Cleveland, OH, USA. [10] Department of Internal Medicine, Division of Medical Oncology, Seoul St. Mary's Hospital, College of Medicine, The Catholic University of Korea, Seoul, South Korea. [11] Department of Hospital Pathology, Seoul St. Mary's Hospital, College of Medicine, The Catholic University of Korea, Seoul, South Korea. [12] Department of Immunology, Mayo Clinic, Jacksonville, FL, USA. [13] These authors contributed equally: Jae-Ho Cheong, Sam C. Wang, Sunho Park. ✉email: jhcheong@yuhs.ac; hwang.taehyun@mayo.edu

Genomic profiling provides prognostic and predictive information about tumor biology that can guide clinical care and improve treatment selection for cancer patients[1–5]. Systemic therapies are an essential component of the treatment regimen for most gastric cancer patients. However, many patients do not derive benefit from the potentially toxic therapies. For example, fluoropyrimidine-platinum doublet is standard of care therapy in the adjuvant setting; however, it only confers an approximate 9% improvement in long-term survival[6]. Thus, biomarkers that predict patient response to chemotherapy are needed to improve treatment precision. We recently reported a single patient classifier system for resectable gastric cancer based on the expression of four genes that is prognostic and predictive of response to fluorouracil-based adjuvant chemotherapy[7]. However, refined risk and treatment stratification tools for gastric cancer are needed to include novel therapies for patients at advanced stage and in the palliative setting, such as immune checkpoint inhibitors.

Recent large-scale next-generation sequencing and molecular profiling efforts have elucidated the genomic landscape of gastric cancer[8,9]. However, the mutational cataloging of cancer genomes fails to incorporate functional analyses of gene-gene interaction and signaling pathway dynamics, and thus risks missing a potentially rich source of prognostic information[10]. We previously developed a machine learning algorithm, NTriPath, that integrates pan-cancer somatic mutation data, gene-gene interaction networks, and pathway databases to identify prognostic cancer-associated molecular pathways[11]. We have used NTriPath to identify prognostic gene signatures for renal cell carcinoma, bladder carcinoma, head and neck squamous cell carcinoma, and melanoma[11].

In this study, we applied NTriPath to identify key gastric adenocarcinoma-specific pathways. Then, we generated gene expression profiles of gastric cancer and used the NTriPath genetic signatures to define four distinct molecular subtypes. We tested the prognostic utility of these genetic signatures and built a molecular subtype-based risk scoring model to predict overall survival and response to chemotherapy and immune checkpoint blockade. Finally, we validated our model in multiple independent cohorts.

## Results

**Identification of a prognostic 32-gene signature and molecular subtypes**. The workflow to identify, test, and validate prognostic and predictive biomarkers in gastric cancer is presented in Fig. 1. The somatic mutation profiles of 6681 patients from 19 different cancer types published by The Cancer Genome Atlas (TCGA) were analyzed (Supplementary Table S1). This data was inputted into NTriPath and pathways that were specifically altered in gastric cancers were identified. To investigate the prognostic utility of these gastric-cancer-specific pathways, we generated microarray-based mRNA expression profiles from pre-treatment tumor samples from 567 patients who underwent resection at Severance Hospital, Yonsei University College of Medicine. 89% of the patients had stage II or III disease and the median duration of follow-up was 61 months (Supplementary Data 1).

We previously found that the top three pathways identified by NTriPath yielded the most prognostic utility[11]. The top three gastric cancer-specific pathways consisted of 32 genes including *TP53*, *BRCA1*, *MSH6*, *PARP1*, and *ACTA2*, which were enriched for DNA damage response, TGF-ß signaling, and cell proliferation pathways (Supplementary Fig. S1, Supplementary Table S2, and Supplementary Data 2). We performed consensus clustering of the 567-patient Yonsei cohort based on the expression level of the 32 genes and found four distinct molecular subtypes based on

consensus cumulative distribution function (CDF) plot and delta area plot as well as manual inspection of the consensus matrices (Fig. 2A and Supplementary Fig. S2). Tumors from Group 1 patients overexpressed genes associated with the cell cycle and DNA repair while cancers from Group 4 patients overexpressed genes found in TGF-β, SMAD, estrogen signaling, and mesenchymal morphogenesis pathways. Tumors from Group 3 patients overexpressed genes found in apoptosis signaling and cell proliferation pathways. Tumors from Group 2 did not show a distinct pattern of overexpressed genes.

In univariate analysis, molecular subtypes correlated significantly with differences in age ($P = 0.003$), stage ($P = 0.021$), Lauren type ($P < 0.001$), and perineural invasion ($P < 0.001$) (Table 1). Patients in Group 1 and Group 2 were older and were more likely to have tumors with intestinal-type histology. Patients in Group 4 had tumors that were more likely to have diffuse-type histology and perineural invasion (Table 1). Finally, significant differences in overall survival were observed between groups; Group 1 patients had the best outcome with the group not reaching median overall survival, while Group 4 patients had the worst outcome with a median overall survival of 65 months (Fig. 2B; $P < 0.001$).

Previous studies showed that tumors associated with Epstein–Barr virus (EBV) infection and microsatellite instability (MSI) are associated with improved patient outcomes[12,13]. We found that there were no differences in the proportion of tumors by that were EBV-positive or MSI (Table 1). Multivariable Cox proportional-hazard analyses using variables that were significant on univariable analysis showed that age, stage, and molecular subtype were independently associated with risk of death (Table 2). To address possible bias resulting from variable selection, we also conducted a regularized Cox regression and found similar results (Supplementary Tables S3 and S4). These findings demonstrate that the 32-gene signature was prognostic independent of clinical and pathologic variables known to correlate with outcomes.

We previously identified 5 transcriptomic-based molecular subtypes that predicted survival[7]. In that report, the inflammatory subgroup had the best prognosis, the mesenchymal subgroup had the worst prognosis, and the other 3 groups had intermediate outcomes and overlapped with each other. We compared the two classification schemes and found that each of the 32-gene groups were generally represented in all 5 molecular subtypes (Supplementary Table S5). There was an enrichment of mesenchymal group patients, who had the worst prognosis, in Group 4, which also had the worst survival. However, even in this situation, only 45.7% of mesenchymal group patients were in Group 4. Finally, when we repeated the multivariate analysis that included the other classification scheme, we found that the 32-gene subtyping continued to associate with overall survival (Supplementary Table S6).

We also compared the 32-gene signature to the classification system published by the Asian Cancer Research Group (ACRG; $n = 300$)[14]. We found that Group 4 (worst survival) was particularly enriched in the MSS/EMT group (worst survival). Group 1 (best survival) was also enriched for MSI (best survival) but 45% of MSI samples were found in Groups 2–4, so it was not a direct correlation. Finally, the MSS/TP53+ and MSS/TP53− groups were even more evenly distributed across Groups 1–4. Thus, the 32-gene signature is not simply a recapitulation of previous classification systems and provides new prognostic information.

**Machine learning to identify a risk score to predict five-year overall survival**. We then sought to leverage the prognostic

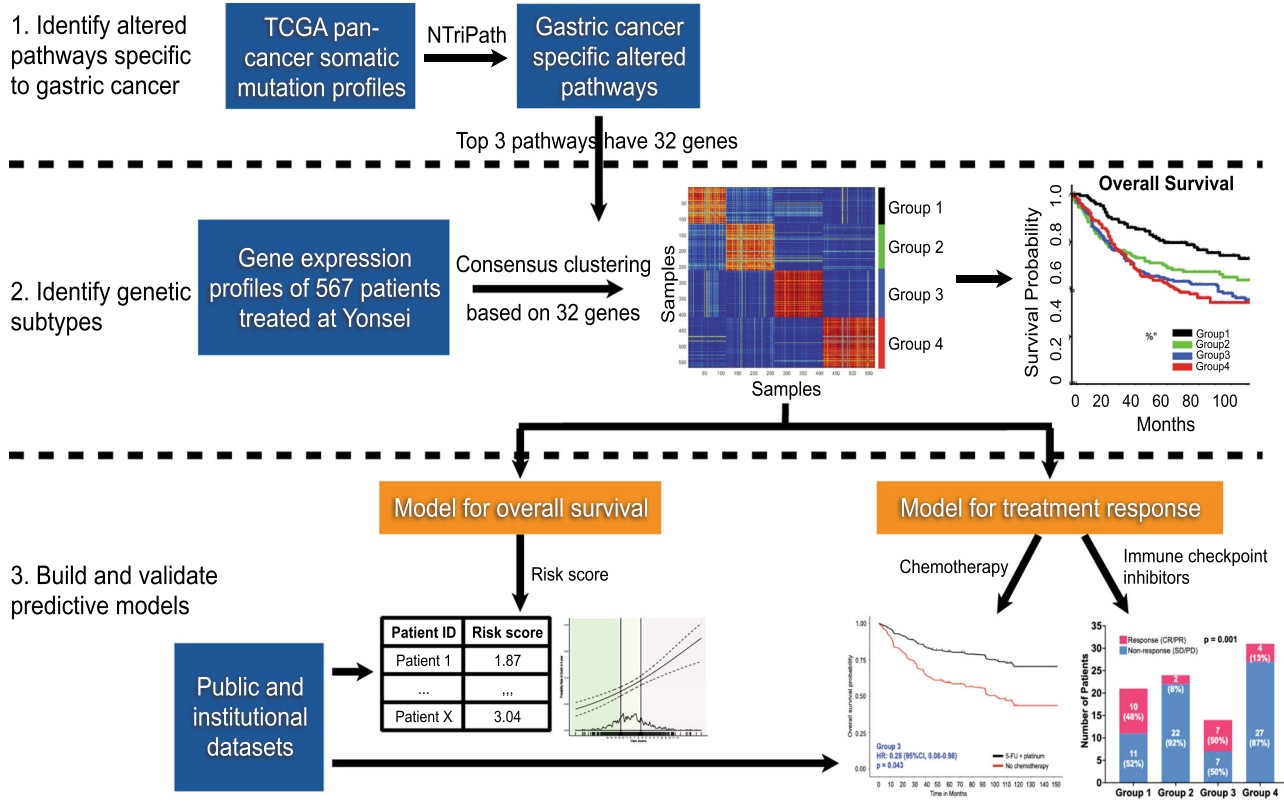

**Fig. 1 Workflow of the current study.** The somatic mutation profiles of 6681 patients from 19 different cancers from TCGA were inputted into NTriPath to identify pathways that were altered specifically in gastric cancer. Microarray-based mRNA expression profiles of 567 gastric cancer patients were generated and inputted into NTriPath. Unsupervised clustering based on the expression of 32 member genes that comprised the top three altered pathways were used to identify molecular subtypes. The prognostic and predictive capability of the molecular subtypes were tested in multiple independent cohorts. TCGA The Cancer Genome Atlas.

capability of the 32-gene signature into a clinically relevant tool that will allow clinicians to estimate the probability of five-year overall survival for gastric cancer patients. Using the Yonsei cohort as a training set, we built a binary classifier using support vector machine (SVM) with linear kernel. The SVM model was trained using Groups 1 and 4. Group 1, which had the best prognosis, was given a negative label and Group 4, which had the worst prognosis, a positive label. We next sought the validate the SVM model using data published by the ACRG (Gene Expression Omnibus: GSE62254)[14], Sohn et al. (n = 267; Gene Expression Omnibus: GSE13861 and GSE26942)[13], and The Cancer Genome Atlas[15]. We found that risk score was prognostic of five-year overall survival (Fig. 2C, Supplementary Fig. S3, Supplementary Data 3–5). Importantly, we found that the risk score was prognostic independent of clinical and pathologic features known to be associated with worse outcomes across all datasets (Table 3 and Supplementary Tables S7–9). These results demonstrated that the machine learning-derived risk score based on the 32-gene signature predicts the probability of five-year overall survival in gastric cancer patients.

**Molecular subtypes predict responses to systemic therapies.** We next investigated whether the molecular subtypes predict response to systemic therapies. The Yonsei cohort included patients who were treated prior to the establishment of adjuvant chemotherapy as standard of care. Thus, we were able to compare patients who underwent surgery alone to patients who received one of three adjuvant chemotherapy regimens: 5-fluorouracil (5-FU) monotherapy, 5-FU and platinum doublet, or 5-FU plus another class of systemic therapy. We performed multivariable

Cox proportional hazard analyses of overall survival and included adjuvant chemotherapy regimens, cancer stage, age, lymphovascular invasion and perineural invasion as covariates, within each genetic subgroup (Supplementary Table S10). We found that the 18 Group 3 patients treated with 5-FU plus platinum showed significantly better overall survival compared to the 28 Group 3 patients who did not receive adjuvant chemotherapy (hazard ratio (HR), 0.28 (95% CI, 0.08–0.96), P = 0.043). In contrast, the 12 Group 1 patients treated with 5-FU plus platinum had worse survival than the 26 Group 1 patients who did not receive adjuvant therapy (HR, 6.80 (95% CI, 1.46–31.6), P = 0.015), (Fig. 3). Receipt of therapy was not associated with survival differences in Group 2 and 4 patients. This data suggests that molecular subtype predicts response to adjuvant chemotherapy.

To determine whether the molecular subtypes also predicted response to immune checkpoint inhibitors, we analyzed samples from patients with recurrent or metastatic gastric cancer treated with immune checkpoint inhibitors. We included 45 patients published by Kim et al.[16] and 45 patients treated at our institutions for a final cohort of 90 patients. Response Evaluation Criteria in Solid Tumors (RECIST) criteria was used to group patients either as responders if they had a complete or partial radiographic response or non-responders if they had stable or progressive disease. The tumor sample from each patient was analyzed with RNA-sequencing. We built the multiclass classifier using SVMs with linear kernel trained with the Yonsei four molecular subtypes. Each patient was classified into one of the four molecular subtypes based on the 32-gene signature by the SVM model. We observed a much higher response rate to pembrolizumab in Group 1 (10 of 21; 48%) and Group 3 (7 of 14;

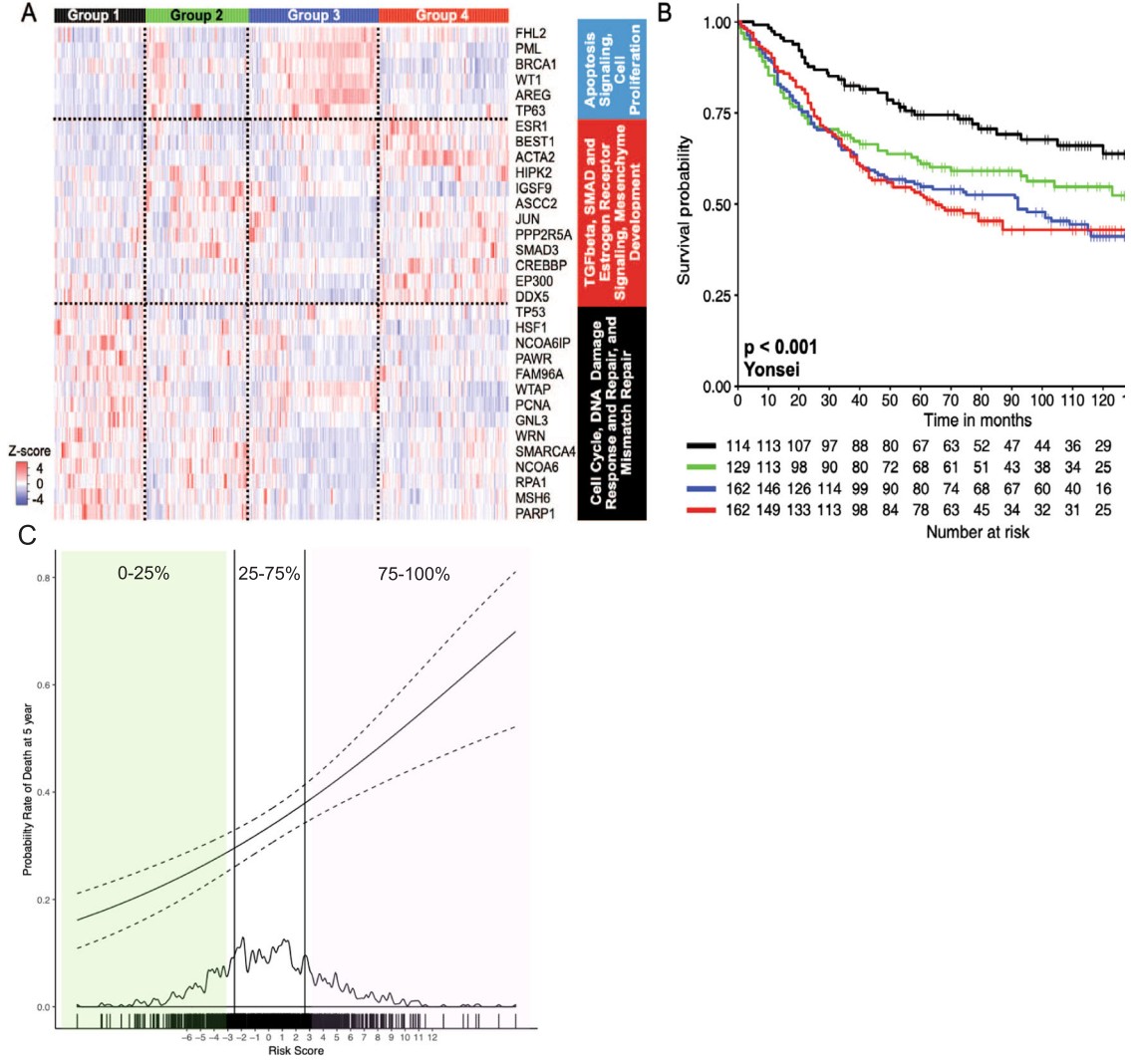

**Fig. 2 The molecular subtypes were prognostic for overall survival. A** Unsupervised consensus clustering using 32-gene signature identified four molecular subtypes in the Yonsei cohort. **B** Kaplan–Meier survival analysis of the four molecular subtypes in the Yonsei cohort. Survival was compared using the log-rank test. **C** The risk score was applied to the Asian Cancer Research Group (ACRG), Sohn et al., and The Cancer Genome Atlas (TCGA) cohorts as a combined group. The dashed curves indicate the 95% confidence interval. The rug plot on top of the x-axis shows the risk score for individual patients. The green region represents patients with scores below the 25th percentile, the white area includes patients with scores from the 25th to 75th percentile, and the purple region includes patients with scores above the 75th percentile. Source data are provided as a source data file.

50%) than in Group 2 (2 of 24; 8%) and Group 5 (4 of 31; 13%) patients ($P = 0.001$; Fig. 4). This showed that molecular subtype also predicts response to immune checkpoint blockade in patients with recurrent or metastatic gastric cancer.

## Discussion

The use of next-generation sequencing has elucidated the genomic landscape of a wide variety of cancers[17]. In select tumor types, mutational profiling provides prognostic and predictive information that guides therapy[1–5]. However, the simple cataloging of somatic mutations fails to capture the downstream effects of gene-gene interactions and altered pathways that play essential functional roles in cancer biology. Thus, we postulated that a potentially rich source of prognostic and predictive information has largely been overlooked. We previously showed that the integration of known molecular interaction networks with somatic mutation data via the machine learning tool we developed, NTriPath, yields prognostic information for renal cell carcinoma, bladder carcinoma, head and neck squamous cell

carcinoma, and melanoma[11]. In this study, we validated this approach for gastric cancer, a disease for which there are few biomarkers available. We generated a molecular subtyping scheme based on the expression of 32 genes that comprised the top three altered pathways specific to gastric cancer. We found that the 32-gene signature predicted both overall survival and response to treatments. Thus, generating the molecular signature from tissue samples obtained at the time of diagnosis may provide clinically important information for prognostication and treatment planning.

We used machine learning to develop a genetic risk score that predicted five-year overall survival. Significantly, the risk score was generated by training on a patient cohort from one of our institutions that allowed for the use of highly granular clinical data, with long-term follow-up. We then validated the risk score in multiple large independent cohorts of gastric cancer patients. The current gold standard for risk stratification of cancer patients is pathologic staging, which is only possible after surgical resection. To complement conventional staging modalities in the preoperative setting, our genetic risk score is an objective measure

**Table 1 Clinical and pathologic variables of the Yonsei cohort as stratified by the four genetic subtypes.**

| Characteristics | Group 1 (Black) | Group 2 (Green) | Group 3 (Blue) | Group 4 (Red) | Total | P value[a] |
|---|---|---|---|---|---|---|
| N | 114 (20.1%) | 129 (22.8%) | 162 (28.6%) | 162 (28.6%) | 567 | |
| Age | | | | | | |
| ≤60 years | 43 (37.7%) | 53 (41.1%) | 90 (55.6%) | 89 (54.9%) | 275 (48.5%) | 0.003 |
| >60 years | 71 (62.3%) | 76 (58.9%) | 72 (44.4%) | 73 (45.1%) | 292 (51.5%) | |
| Sex | | | | | | |
| Male | 77 (67.5%) | 93 (72.1%) | 114 (70.4%) | 102 (63.0%) | 386 (68.1%) | 0.346 |
| Female | 37 (32.5%) | 36 (27.9%) | 48 (29.6%) | 60 (37.0%) | 181 (31.9%) | |
| Stage | | | | | | |
| I | 10 (8.8%) | 3 (2.3%) | 7 (4.3%) | 1 (0.6%) | 21 (3.7%) | 0.021 |
| II | 33 (28.9%) | 36 (27.9%) | 33 (20.4%) | 45 (27.8%) | 147 (25.9%) | |
| III | 69 (60.5%) | 87 (67.4%) | 115 (71.0%) | 108 (66.7%) | 379 (66.8%) | |
| IV | 2 (1.8%) | 3 (2.3%) | 7 (4.3%) | 8 (4.9%) | 20 (3.5%) | |
| Tumor location | | | | | | |
| Antrum | 60 (52.6%) | 83 (64.3%) | 85 (52.5%) | 88 (54.3%) | 316 (55.7%) | 0.338 |
| Body | 38 (33.3%) | 33 (25.6%) | 57 (35.2%) | 54 (33.3%) | 182 (32.1%) | |
| Cardia | 4 (3.5%) | 11 (8.5%) | 16 (9.9%) | 13 (8.0%) | 44 (7.8%) | |
| Whole | 1 (0.9%) | 0 (0.0%) | 2 (1.2%) | 3 (1.9%) | 6 (1.1%) | |
| Missing | 11 (9.6%) | 2 (1.6%) | 2 (1.2%) | 4 (2.5%) | 19 (3.4%) | |
| Lauren type | | | | | | |
| Diffuse | 21 (18.4%) | 33 (25.6%) | 55 (34.0%) | 89 (54.9%) | 198 (34.9%) | <0.001 |
| Intestinal | 56 (49.1%) | 60 (46.5%) | 35 (21.6%) | 43 (26.5%) | 194 (34.2%) | |
| Mixed | 6 (5.3%) | 9 (7.0%) | 4 (2.5%) | 6 (3.7%) | 25 (4.4%) | |
| Other | 31 (27.2%) | 26 (20.2%) | 68 (42.0%) | 24 (14.8%) | 149 (26.3%) | |
| Missing | 0 (0.0%) | 1 (0.8%) | 0 (0.0%) | 0 (0.0%) | 1 (0.2%) | |
| Lymphovascular invasion | | | | | | |
| Positive | 57 (50.0%) | 70 (54.3%) | 65 (40.1%) | 76 (46.9%) | 268 (47.3%) | 0.094 |
| Negative | 56 (49.1%) | 57 (44.2%) | 95 (58.6%) | 86 (53.1%) | 294 (51.9%) | |
| Missing | 1 (0.9%) | 2 (1.6%) | 2 (1.2%) | 0 (0.0%) | 5 (0.9%) | |
| Perineural invasion | | | | | | |
| Positive | 19 (16.7%) | 17 (13.2%) | 28 (17.3%) | 63 (38.9%) | 127 (22.4%) | <0.001 |
| Negative | 92 (80.7%) | 109 (84.5%) | 132 (81.5%) | 99 (61.1%) | 432 (76.2%) | |
| Missing | 3 (2.6%) | 3 (2.3%) | 2 (1.2%) | 0 (0.0%) | 8 (1.4%) | |
| Epstein–Barr Virus | | | | | | |
| Positive | 4 (3.5%) | 2 (1.6%) | 7 (4.3%) | 6 (3.7%) | 19 (3.4%) | 0.568 |
| Negative | 44 (38.6%) | 51 (39.5%) | 60 (37.0%) | 55 (34.0%) | 210 (37.0%) | |
| Missing | 66 (57.9%) | 76 (58.9%) | 95 (58.6%) | 101 (62.3%) | 338 (59.6%) | |
| Microsatellite instability | | | | | | |
| Yes | 10 (28.6%) | 7 (14.0%) | 2 (11.8%) | 2 (2.7%) | 21 | 0.36 |
| No | 25 (71.4%) | 43 (86.0%) | 15 (86.7%) | 73 (97.3%) | 156 | |
| Chemotherapy receipt | | | | | | |
| Yes | 85 (74.6%) | 104 (80.6%) | 129 (79.6%) | 135 (83.3%) | 453 (79.9%) | 0.36 |
| No | 28 (24.6%) | 24 (18.6%) | 32 (19.8%) | 26 (16.0%) | 110 (19.4%) | |
| Missing | 1 (0.9%) | 1 (0.8%) | 1 (0.6%) | 1 (0.6%) | 4 (0.7%) | |

[a]P values were calculated using the Chi-square test.

to estimate risk and represents a powerful new risk stratification tool that has the potential to optimize treatment choice and sequence, on an individualized basis. Our promising results should be validated in a prospective manner.

There are currently few predictive biomarkers to guide treatment choices for gastric cancer patients. Patients with gastric cancers with HER2 amplification benefit from trastuzumab, but this is only applicable for a small proportion of patients[18]. Similarly, while Pietrantonio et al. recently showed that patients with resectable gastric cancer with MSI tumors did not benefit from chemotherapy as compared to surgery alone[19], only 5–10% of gastric cancers are MSI[8]. In terms of checkpoint inhibitors, therapy is most often guided by MSI status and PD-L1 expression, but their predictive capability is modest[20]. For example, the KEYNOTE-061 study found that pembrolizumab did not improve the overall survival of gastric cancer patients, including patients whose tumors had high PD-L1 expression[21]. Our 32-gene assay has the potential to augment currently available biomarkers and improve the precision of gastric cancer care. We

found that Group 3 patients were the only ones who demonstrated improved survival associated with adjuvant 5-FU and platinum-based chemotherapy, while Group 1 patients had worse survival associated with adjuvant 5-FU and platinum. Our findings are potentially widely applicable as fluoropyridine and platinum-based regimens are first-line therapy for patients with regional and metastatic disease, with which most patients present, especially in the West[6,12,22,23].

Significantly, we also found that the molecular classification is associated with response to immune checkpoint inhibitors as Groups 1 and 3 patients demonstrated significantly higher response rates than patients in Groups 2 and 4. Importantly, we did not find that MSI-H tumors were overrepresented in Groups 1 and 3. Intriguingly, Group 3 patients demonstrated a good response rate to both 5-FU and platinum doublet chemotherapy as well as anti-PD-1 treatments. Consideration may be given to a clinical trial combining all three agents in this patient population. Meanwhile, even though Group 1 patients had the best prognosis, the use of 5-FU and platinum therapy was associated with worse

**Table 2 Multivariable analysis of the Yonsei cohort.**

| Characteristics | Hazard ratio (95% CI) | P value[a] |
|---|---|---|
| Age | | |
| ≤60 years | Reference | |
| >60 years | 1.95 (1.51, 2.51) | <0.001 |
| Stage | | |
| I | Reference | |
| II | 1.86 (0.66, 5.25) | 0.241 |
| III | 3.58 (1.31, 9.75) | 0.013 |
| IV | 18.2 (6.08, 54.5) | <0.001 |
| Tumor location | | |
| Antrum | Reference | |
| Body | 1.02 (0.78, 1.34) | 0.871 |
| Cardia | 0.90 (0.56, 1.45) | 0.671 |
| Whole | 1.49 (0.60, 3.72) | 0.395 |
| Lauren type | | |
| Diffuse | Reference | |
| Intestinal | 0.89 (0.65, 1.22) | 0.481 |
| Mixed | 0.71 (0.35, 1.42) | 0.332 |
| Other | 1.20 (0.86, 1.68) | 0.294 |
| Perineural invasion | | |
| Negative | Reference | |
| Positive | 1.12 (0.81, 1.55) | 0.491 |
| Molecular subtype | | |
| Group 1 | Reference | |
| Group 2 | 1.97 (1.31, 2.95) | 0.001 |
| Group 3 | 1.73 (1.12, 2.65) | 0.013 |
| Group 4 | 2.18 (1.44, 3.31) | <0.001 |

[a]P value for the interaction term is based on the Cox proportional hazards model.

**Table 3 Multivariable analysis of risk score for the combined TCGA, ACRG, and Sohn et al. cohorts.**

| Characteristics | Hazard ratio (95% CI) | P value[a] |
|---|---|---|
| Age | | |
| ≤60 years | Reference | |
| >60 years | 1.89 (1.53, 2.32) | <0.001 |
| Stage | | |
| I | Reference | |
| II | 1.33 (0.88, 2.01) | 0.18 |
| III | 2.65 (1.82, 3.88) | <0.001 |
| IV | 5.31 (3.60, 7.83) | <0.001 |
| Risk Score (per unit increase) | 1.06 (1.04, 1.09) | <0.001 |

[a]P value for the interaction term is based on the Cox proportional hazards model.

outcomes. For these patients, consideration may be given to treatment with immune checkpoint blockade, to which these patients demonstrated significant response.

Our findings build upon our previous work that identified a genetic classifier that was prognostic and predictive of response to adjuvant chemotherapy in stage II and III patients[7]. Thus, our previous study and current report demonstrate that comprehensive genomic profiling of gastric cancer can provide clinically impactful information to guide therapy. However, our current study is limited by the retrospective nature of our analysis that may be confounded by selection bias. For example, our training set was based on samples obtained before adjuvant chemotherapy was standard of care. Thus, the decision to refer a patient for post-operative chemotherapy versus observation are unknown. Future validation studies of the utility of the 32-gene signature to predict response to both 5-FU and platinum therapy and immune checkpoint inhibitors are needed to confirm our findings, which are based on modest-sized cohorts. Ideally, these future studies

are performed in a prospective fashion. Next, while we showed that the molecular classifier was associated with increased response to immune checkpoint inhibitors, response may not accurately predict overall survival. Thus, the prognostic and predictive utility of the molecular classification scheme should be validated prospectively using the most clinically relevant end-points, such as overall survival and/or quality of life. Finally, the molecular mechanisms underpinning the prognostic and predictive power of the 32-gene signature are important topics for future study.

## Methods

**Patient cohorts.** The study was approved by the Institutional Review Board of the College of Medicine at Yonsei University and the Catholic University of Korea. We analyzed samples from 567 gastric adenocarcinoma patients who underwent surgical resection at Yonsei University (Seoul, Korea) from 1999 to 2010, 28 patients treated at Seoul St. Mary's Hospital (Seoul, Korea) from 2018 to 2020, and 17 patients from Yonsei (Seoul, Korea) from 2014 to 2017. We also examined data from cohorts previously published by The Cancer Genome Atlas Project (TCGA)[15], Asian Cancer Research Group (ACRG)[9], Sohn et al.[13], and Kim et al.[16].

**NTriPath identifies gastric cancer-associated pathways.** The bioinformatics tool NTriPath uses somatic mutation profiles, pathway databases, and gene-gene interaction networks to identify cancer-type-specific associated pathways[11]. We inputted somatic mutation data of 6681 patients across 19 cancer types from the TCGA via cBioPortal on October 24th 2014[24]. We constructed a gene-gene interaction network by combining networks described by Zhang et al.[25]. We used 4620 functional modules from human gene-gene interaction networks as a pathway database[26]. NTriPath yields cancer type and pathway associations that can be used to identify cancer type-specific altered pathways. In the exploratory study with the Yonsei cohort, we conducted a pathway analysis using gene set enrichment analysis to identify gastric cancer-specific biological pathways that are enriched in a list of genes selected by NTriPath[27]. Statistical analysis was performed with PANTHER (Protein ANalysis THrough Evolutionary Relationships) Classification System using Fisher exact test with false discovery rate (FDR) < 0.05 (http://www.pantherdb.org) for gene set enrichment[28].

To measure the statistical significance of gastric cancer type and pathway associations, we performed a permutation test where we randomly permuted somatic mutation data and repeated experiments 1000 times to calculate empirical $p$ values. We defined gastric cancer type-specific altered pathways based on the following strict criteria: (1) pathways must be ranked within the top Kth compared to other pathways in each cancer type based on their association scores in matrix S and (2) pathways must have significant Benjamini–Hochberg adjusted $P$ values using an FDR cutoff of 0.05. In this work, we used the top 5 ranked pathways that had an FDR-adjusted $P$ value < 0.05 to identify gastric cancer-specific altered pathways.

NTriPath source code with input datasets are available at the following link: https://github.com/hwanglab/NTriPath.

**Preprocessing of gene expression data.** We generated gene expression profiles of 567 gastric cancer patients treated at Yonsei University using Illumina Human-6 V2 Expression BeadChips. Raw microarray data were transformed to the log2 base scale and then were preprocessed by quantile normalization using quantilenorm function in MATLAB R2018b. The expression level of each gene was obtained by averaging multiple probes from the same gene. Genes that had missing values were excluded. This processed data was used to perform consensus clustering to identify patient subgroups.

We downloaded the datasets published by the ACRG[9] (GSE62254) and Sohn et al.[13] (GSE13861 and GSE26942) through the GEO database. The ACRG data was generated by Affymetrix Human Genome U133 Plus 2.0 Array and was preprocessed by Robust Multi-array Average (RMA) with Affymetrix Power Tools package using Affymetrix default analysis settings. We downloaded the preprocessed data and generated the expression level of each gene by averaging multiple probes from the same gene. The Sohn et al.[13] data was generated by Illumina HumanHT-12 V3.0 beadchip. Raw microarray data were transformed to the log2 base scale and then were preprocessed by quantile normalization using quantilenorm function in MATLAB R2018b. Processed gene expression profiles from the TCGA stomach adenocarcinoma (STAD) group were downloaded via cBioPortal.

The raw sequencing data for the Kim et al.[16] cohort was accessed via the European Nucleotide Archive (accession number PRJEB25780). For patients treated at Seoul St Mary's Hospital, we used Illumina TrueSeq RNA Exome library kit and generated 2x150 bp paired-end reads. In the datasets, the mean insert length is 125 base pairs. An average of 143.7 million paired-end reads were sequenced per patient. We then processed the Kim et al.[16] and St. Mary's datasets. After Illumina universal adapters were trimmed, reads with lower quality bases or noninformative reads were filtered out by Fastp v0.20.1 (with the input parameters:

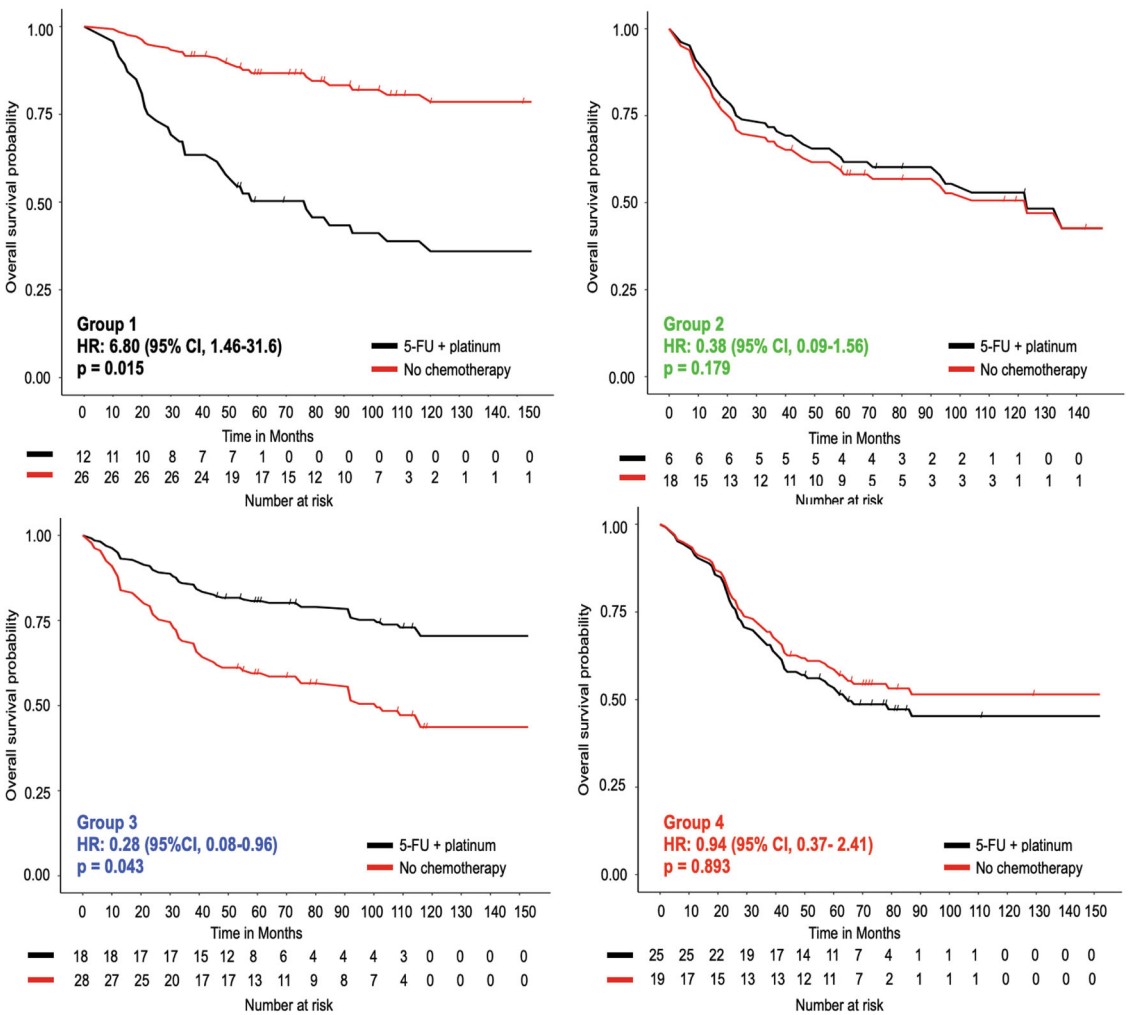

**Fig. 3 The molecular subtypes are associated with response to adjuvant 5-fluorouracil (5-FU) and platinum chemotherapy.** Kaplan–Meier curves for overall survival for patients treated at Yonsei University, stratified by molecular subtype. Patients who underwent surgery with no adjuvant chemotherapy are compared to ones who received surgery and adjuvant 5-FU and platinum. The log-rank test was used to test statistical significance. Source data are provided as a source data file.

'-y -c'). The RNA-seq reads were converted to transcriptome abundant matrix in the format of TPM (Transcripts Per Kilobase Million) via Salmon v0.14.1 (with the input parameters: '--validateMappings --libType A') The Salmon reference genome sequence index was built on human reference transcript sequences (GRCm37.v19) with the gene model, Gencode v19.

**Unsupervised consensus clustering using top-ranked gastric cancer-associated pathways**. To investigate the prognostic utility of top-ranked gastric cancer-associated pathway signatures generated by NTriPath, we performed consensus clustering using non-negative matrix factorization (NMF). This was performed with the expression profiles of the 567 Yonsei cohort patients and the member genes of top-ranked gastric cancer-associated pathways with various numbers of subgroups ($k = 2–7$).

We first performed consensus NMF clustering based on expression of the 32 member genes of the top three ranked gastric cancer-associated pathways. We identified four subgroups by visual inspection and consideration of the cophenetic correlation coefficient, the cumulative distribution function (CDF) 6 of consensus matrix and the Delta values for each $k$ group (Supplementary Fig. S2A–C).

To test the robustness of NMF subgroups, we built a multiclass classifier from four NMF subgroups using SVMs with linear kernel and performed leave-one-out cross-validation to predict NMF subgroup membership[29,30]. Specifically, we used SVMs with linear kernel using the "all-pair comparisons" approach, where we build $\frac{K(K-1)}{2}$ binary classifiers and each classifier is trained to distinguish each pair of classes. To obtain the class membership probabilities (a vector of length $K$) from the prediction results (a vector of length $\frac{K(K-1)}{2}$) of the binary classifiers, we used the aggregation method proposed by Park et al.[11]. In order to prevent overfitting, we used a simpler model: the aggregating weights (a vector of length) assigned on the SVM classifiers in the aggregation method were fixed to a uniform vector under

the assumption that each binary classification problem is equally important; and a regularization parameter, in each binary classification the problem was set to 1 (equal to the default setting in LibSVM toolbox (Ver 3.17). We measured the performance of the multiclass classifier by generating receiver operating characteristic curves and measuring the area under curve (AUC). The mean AUC for the classification was 0.981. We used this multiclass classifier using SVMs with linear kernel trained with the Yonsei four subtypes to classify patients from other cohorts into the four subtypes in the supervised classification setting.

**Predictive model for risk scores to predict overall survival**. SVM using linear kernel was used to build a predictive model to generate risk scores to predict overall survival following surgical resection of gastric cancer. We built the predictive model based on SVM using linear kernel from the Yonsei cohort and tested the model in the cohorts published by ACRG[9], Sohn et al.[13], and TCGA[15].

Since gene expression profiles from cohorts were generated by different microarray and illumine sequencing platforms, we used ComBat (http://www.bu.edu/jlab/wp-assets/ComBat) to remove any potential batch effects that may result from using datasets from different platforms and/or experiments and used the processed datasets for further analysis[31]. We first selected patients from Groups 1 and 4 from the Yonsei cohort, which had the best and worst outcomes, respectively. We assigned "−" and "+" labels to each subgroup, respectively. We built SVM with a linear kernel function using the 32 gene expression profiles of patients from Groups 1 and 4 with the +/− label information.

We applied the trained SVM to the same gene expression profiles of patients from the ACRG[9], Sohn et al.[13], and TCGA[15] cohorts. A SVM outputs a continuous risk score for each test sample. A larger positive score value indicates the higher degree of confidence that the sample belongs to the positive class (i.e., Group 4, poor prognosis group). Similarly, a smaller negative score value indicates a higher degree of confidence that the sample belongs to the negative class (i.e., Group 1,

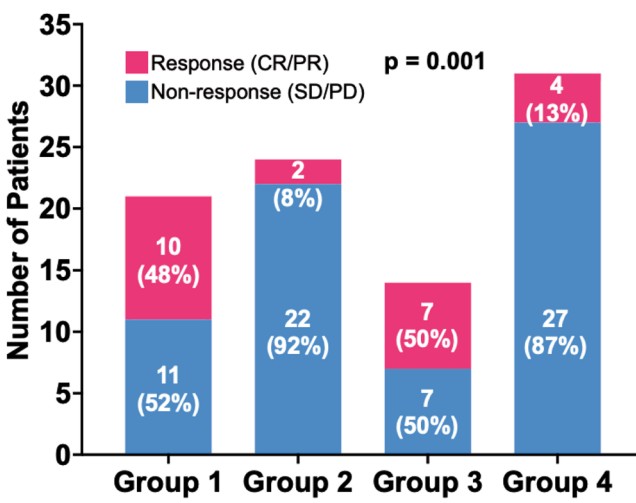

**Fig. 4 The molecular subtypes are associated with response to immune checkpoint inhibitors.** Patients with advanced gastric cancer who were treated with immune checkpoint blockade were stratified by molecular subtypes. Response is defined by complete response (CR) or partial response (PR). Non-response is defined as stable disease (SD) or progressive disease (PD). The chi-square test was used to compare groups. Source data are provided as a source data file.

good prognosis group). Thus, the risk score values of test samples can be used to predict survival probability.

The source code and datasets for risk score prediction are available at the following link: https://github.com/hwanglab/NTriPath.

**Statistical analysis.** For univariate analyses, we used chi-squared (categorical variables) or ANOVA tests (continuous variables) to determine whether the genetic subtype variable was correlated with other clinical variables. For multivariate analyses, we employed a Cox proportional hazards model to relate patients' survival time with subtype and other covariates. The clinical variables identified as statistically significant by univariate analysis were selected as the covariates in the multivariate analyses.

Since the results from the multivariate analysis could be biased due to the selection step using the univariate analysis, we also conducted a regularized Cox regression (Cox model with lasso regularization, implemented with the glmnet package in R). Using the Yonsei cohort, we set all clinical variables (except the variables with a high proportion of missing data, such as EBV, which was missing in ~60% of samples) as predictors and let the model select the optimal features for survival. We also calculated the 95% confidence interval of the hazard ratio and the $p$ value of each variable using bootstrapping (the null hypothesis H_0 is that each coefficient $\beta_j = 0$). We ran the model 10,000 times in the bootstrapping setting (the training data in each run was resampled with replacement) and calculated the hazard ratio, 95% confidence interval, and the $p$ value of each coefficient based on the empirical distribution of the estimator of the coefficient.

When generating each Kaplan–Meier (KM) plot, we used the log-rank test to compare the survival curves. To generate the adjusted KM curves, we applied a Cox proportional hazards model to each subgroup and averaged the predicted survival curves by the treatment variable (i.e., 5-FU + platinum, 5-FU alone, or none) in that subgroup. All analyses were done in R (version 4.0.3) and MATLAB R2018a.

**Reporting summary.** Further information on research design is available in the Nature Research Reporting Summary linked to this article.

## Data availability

Gene expression profiles of patients treated at Yonesi University can be found here: [https://www.ncbi.nlm.nih.gov/geo/query/acc.cgi?acc=GSE183136] and [https://www.ncbi.nlm.nih.gov/geo/query/acc.cgi?acc=GSE84437]. RNA-sequencing data for patients treated with immune checkpoint inhibitors is available in the European Genome-Phenome Archive under the Dataset ID EGAD00001008091: [https://ega-archive.org/studies/EGAS00001005588]. The ACRG data file is available here: [https://www.ncbi.nlm.nih.gov/geo/query/acc.cgi?acc=GSE62254]. Data from the Sohn et al. cohort is available here: [https://www.ncbi.nlm.nih.gov/geo/query/acc.cgi] and [https://www.ncbi.nlm.nih.gov/geo/query/acc.cgi]. Data from the Kim et al cohort is available

here: [https://www.ebi.ac.uk/ena/browser/view/PRJEB25780]. Source data are provided with this paper.

## Code availability

The MATLAB and R scripts used for this study are available here: https://github.com/hwanglab/Yonsei_gastric_cancer_32genes. The DOI for the code is https://doi.org/10.5281/zenodo.5779446 [https://zenodo.org/record/5779446#.YbtuSH3ML0o][32].

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

## Acknowledgements

This research was supported by a grant from the Korean Health Technology R&D Project through the Korean Health Industry Development Institute (KHIDI), funded by the Ministry of Health & Welfare, Republic of Korea (grant No. HI13C2162). SCW is a Disease Oriented Clinical Scholar at UT Southwestern and supported by the NCI/NIH (K08 CA222611). MRP is a Dedman Family Scholar in Clinical Care. JDK holds a Physician-Scientist Institutional Award from the Burroughs Wellcome Fund (award no. 1018897).

## Author contributions

J.C., S.C.W., M.R.P., W.J.H., S.H.N., T.H.H. conceptualized the project. J.C., S.C.W., S.P., T.H.H. contributed to data curation. J.C., S.C.W., A.L.C., H.Z., B.H., C.H., T.H.H. contributed to formal analysis. J.C., T.H.H. contributed to funding acquisition. J.C., S.P., H.K., T.H.H. contributed to project investigation. J.C., S.P., T.H.H. contributed to methodology development. J.C., H.K., H.S.K., W.J.H., S.H.N., I.K., S.H.L., T.H.H. contributed to resource acquisition. S.P., T.H.H. contributed to software development. J.C., T.H.H. contributed to project administration. J.C., T.H.H. supervised the project. J.C., S.C.W., S.P., M.R.P., T.H.H. contributed to the writing of the original draft. J.C., S.C.W., S.P., M.R.P., A.L.C., H.K., H.Z., W.J.H., S.H.N., B.H., C.H., J.D.K., I.K., S.H.L., T.H.H. contributed to reviewing and editing the final draft.

## Competing interests

J.C., S.C.W., S.P., S.H.L., and T.H.H. are co-inventors of a patent for the 32 gene signature which is currently under consideration (PCT/KR2021/018966). All other authors declare no competing interests.
