## [Peer Review File · Nature Communications]

REVIEWER COMMENTS

Reviewer #1 (Remarks to the Author): Expert in gastric cancer and prognosis

The authors identified a 32-gene signature that categorized gastric cancer patients into four subtypes that were independently associated with overall survival and predicted response to chemotherapy and PD-1 inhibitors. It is an interesting study but lack of innovation. Moreover, there are some problems in the design of this study.

1. The gene signature was established from 19 different cancers and not gastric cancer specific.
2. The number of patients who received immune checkpoint Inhibitors was too small to get a convincing result. For example, there is only 2 patients in group C of figure 5B.
3. Patients in group 1 had the best prognosis, could not benefit from adjuvant chemotherapy of 5-fluorouracil and platinum and have higher response rate to immune checkpoint Inhibitors. It seems that patients in group 1 share similar features to MSI-H patients. The author should explain the difference between group 1 and MSI-H.
4. The survival difference between patients in group 3 and 4 is not significant. The survival curve overlapped in the first 30 months for patients in group 2, 3 and 4.
5. The authors should explain the relationship between the clinical features and the classification of the gene signature. The percentage of stage IV patients was lowest in group 1 which may explain the best prognosis of group 1.

Minor points

There is no figure legend for figure 5.

Reviewer #2 (Remarks to the Author):Expert in gastric cancer and prognosis

By applying NTriPath algorithm, investigators identified signaling pathways specific to gastric cancer and 32 genes associated with top 3 signaling pathways. Authors further identified four subtype by using unsupervised clustering approach with expression data of 32 genes. Four subtypes are not only associated with overall survival of patients but also shows good correlation with benefit of standard treatments such as chemotherapy and immunotherapy.

Overall, this is well designed and executed study for finding and validating novel prognostic subtypes. However, there are some issues that need to be resolved before publication.

Major issues

1. Selection process for 32 genes is not clearly described. What were criteria for selecting 32 genes and cutoff for selection?
2. Authors developed two stratification methods, 4 subtypes by clustering and 3 subtypes by generating risk score. It is not clear why they need to generate two different stratifications.
3. Authors used clustering for validation of subtypes in independent cohorts. Clustering is not ideal approach for validation. More reliable algorithm like pamr or similar one need to be used.
4. Concordance with previously identified subtypes are not addressed. Previous studies including one done by authors identified 4 or 5 subtypes. Similarity or dissimilarity of current subtypes with previous ones should be directly compared as authors used same data sets.

Reviewer #3 (Remarks to the Author): Expert in machine learning

The authors propose and study a 32-gene signature for prognosis and prediction of response to therapy in gastric cancer. The approach builds on previous work (Park et al., Bioinformatics 2016; ref #11 in the current ms) that put forward a tool (NTriPath) aimed at identifying pathways associated with survival. The main novelty of the current manuscript lies in the application to gastric cancer.

The general workflow is as follows. Using large pan-cancer data (from TCGA) the previously established NTriPath is used to identify a gastric cancer-specific signature whose association with prognosis and therapeutic outcome is studied using clinical data.

The paper is well motivated and addresses an interesting question. It is nice to see work with a specific clinical focus and the authors have made a good effort to bring together several clinical datasets.

However, a number of questions concerning the data analysis should be addressed and clarified, as detailed below, and potential limitations stated clearly. This is particularly important in this domain, as care is needed in the use and interpretation of machine learning methods to inform clinical practice.

Main points:

1. Clarify dataset use for analysis steps. The manuscript as written is not fully clear with respect to how various datasets were used for different steps in the analysis. Please see detailed comments below for specific examples with line-by-line references, but in general it would be important throughout the paper to make this information absolutely as clear as possible. Each of the steps (gene selection, clustering, learning prediction models etc.) involves learning from data in some form and is important to be clear about what parameters were trained on which data and which parameters were then carried across to other datasets.

2. Selection bias. The results concerning prediction of response to therapy are based on non-randomized data (lines 212-216). This may result in bias relevant to the results presented. The authors note this point in Discussion (line 312), but potential bias and its importance for clinical interpretation should be made clear throughout. You note that patients were included who were treated prior to adjuvant therapy becoming standard of care. If known, can you say a little more about how treatment decisions were made, i.e. based on which factors? This may help readers to think about potential selection bias.

3. Clarify small-sample issues. In places, the models appear to be used in very small sample settings. For example, in the immune checkpoint analysis (line 230 onwards), n=45 patients were available. From the current description, it appears that a multi-class SVM was trained using these data. If so, this is a very small number with which to train and test a classifier. Furthermore, this includes the four subtypes, hence the subtype-specific sample sizes must be even smaller. Please clarify this point and report numbers of patients in the main text itself and not just percentages.

Detailed comments:

4. Abstract. This should be edited in light of points 1-3. In particular please note the potential for selection bias and clarify the role of initial and validation datasets.

5. Line 143: please clarify precisely which expression data were used for the consensus clustering.

6. Line 164. The multivariable Cox analysis uses "variables that were significant on univariable analysis". What data were the univariate and multivariable models respectively fit on? If the same dataset for both, then are the latter estimates biased (due to the selection step)?

7. Line 170 onwards, subsection "Validation of the 32-gene Prognostic Signature to Discover molecular subtypes". It is stated: "Unsupervised consensus clustering using the 32-gene signature again identified four molecular subtypes". Was the clustering re-done using the ACRG and Sohn et al. datasets in the sense that consensus clustering was performed using the 32 genes previously

identified, but using expression data from these two studies? If so, it should be noted that a new clustering using the same genes will not in general give the same partition of the feature space, i.e. the clusters in the validation data may occupy different regions of gene expression space than in the original, discovery dataset. Furthermore, can you explain how the clusters were matched up, i.e. how did you match a specific cluster discovered in ACRG, say, with a cluster in the original data?

8. SVM learning, Line 193. It is stated that a SVM was trained "to estimate five-year overall survival." However in line 195 it is stated that a binary classifier was learned, using labels for groups 1,4 as training labels. Is the idea to use these labels to focus on extremes, rather than directly modelling survival? Later on it is stated that using the validation data (ACRG and Sohn et al.) "the risk score, as a continuous variable, was prognostic of five-year overall survival". It is not clear what is meant here: can you rephrase?

9. Added value compared with clinical features, line 205. In the validation data, can you compare the fitted SVM with a regression using only the clinical and pathological features? This would help to directly quantify added value in terms of held-out classification/prediction accuracy.

10. Line 212. It would be relevant to point out the possibility of selection bias here (see #2 above).

11. Line 224. Please state absolute numbers of patients and not only HR.

12. Line 230. Using the Kim et al. data how were the patients clustered into groups 1-4 and how were the clusters matched with those from Yonsei? It is stated "There were 45 patients with RNA-seq data available for analysis. Using that data, we built a multi-class SVM..." Does this mean an SVM was trained on n=45 data covering 4 classes? How was this done? A sample size of n=45 is very small to train and test a classifier.

14. Line 237. Please state patient numbers and not only proportions. With n=45 over four groups, how many patients were in each group?

15. Line 243. Please address the same questions in #14,15 also for the n=28 Seoul St Mary's Hospital data.

16. Discussion. Can you state how you would expect the models to be used in practice? What would be the clinical workflow? What next steps would be needed to take these ideas forward? This would be very useful to contextualise the work and give readers a clear sense of the scope.

17. Gene-set analysis, line 341. Please state what statistical test was used within gene-set analysis.

We thank the reviewers for their comments and suggestions. Based on the comments from the reviewers, it was apparent to us that there were instances where we failed to clearly articulate our methodology, leading to confusion. This was exacerbated by the inclusion of data that was only tangentially related to the focus on the paper. Thus, for the re-submission, we chose to remove some figures to improve the focus of the manuscript. This is detailed below. In addition, we have also included more samples from patients treated with immune checkpoint blockade. Finally, we have made every effort to address all of the points raised by the 3 reviewers. Guided by their comments and suggestions, we believe the manuscript is significantly improved. Please see a detailed response to each point below. Please note that in the revised manuscript, we have designated new or changed text in red.

Reviewer #1 (Remarks to the Author): Expert in gastric cancer and prognosis

The authors identified a 32-gene signature that categorized gastric cancer patients into four subtypes that were independently associated with overall survival and predicted response to chemotherapy and PD-1 inhibitors. It is an interesting study but lack of innovation. Moreover, there are some problems in the design of this study.

1. The gene signature was established from 19 different cancers and not gastric cancer specific.

We apologize that our methods description was not clear in the original submission. To clarify, we developed NTriPath by analyzing somatic mutations in cancer patients with 19 different types of tumors to identify altered gene signatures that are cancer-type specific (Park et al, Bioinformatics, 2016, PMID 26635139). In our initial report of NTriPath, we established kidney-specific, bladder-specific, head and neck cancer-specific, and skin cancer-specific signatures. For this report, our NTriPath output was a gastric cancer-specific signature. We have made changes to the Introduction and Methods sections of the manuscript to clarify this point.

2. The number of patients who received immune checkpoint inhibitors was too small to get a convincing result. For example, there is only 2 patients in group C of figure 5B.

Unfortunately, transcriptomic data that is annotated with gastric cancer patient response to immunotherapy is currently rare. To our knowledge, there is currently only one published dataset that has RNA-seq data from gastric cancer patients treated with immune checkpoint inhibitors, which is the report by Kim et al (Nature Medicine, 2018, PMID 30013197). In that report, they included 45 patients.

For the original submission, we first analyzed the RNA-seq data for these patients and classified each patient into one of four molecular subtypes. Then we evaluated the response rate of each group to immune checkpoint inhibitors. These findings were presented as the original Figure 5A (shown below). We then analyzed 28 patients from our group’s work; this was shown in the original Figure 5B (shown below). In the original report, we presented the two groups as separate cohorts, which significantly reduced the number of patients in each subgroup, as pointed out by the reviewer.

For the resubmission, we were able to add another 17 patients from our own group to the original 28, for a total of 45 patients, that now matches the largest publicly available dataset from the Kim et al. study. Thus, we have now analyzed a total of 90 patients treated with immune checkpoint inhibitors. For the resubmission, we pooled all 90 patients to maximize the number of patients in each subtype. As shown in the new figure (Figure 4 in the updated manuscript) shown below, Group 1 had an overall response rate (ORR) to immune checkpoint inhibitors of 48% (N=21) and Group 3 had an ORR of 50% (N=14). An ORR was observed in only 8% of Group 2 patients (N=24) and 13% of Group 4 patients (N=31). This difference was statistically significant ($p = 0.001$). Overall, we believe this data shows that the molecular subtypes we identified are predictive of response to immune checkpoint blockade.

Figure 4

Of note, recent randomized control trials have shown that the ORR observed for advanced gastric cancer patients treated with immune checkpoint inhibitors is less than 20% (12% in KEYNOTE-059 (Fuchs et al, JAMA Onc, 2018; PMID: 29543932)), 16% in KEYNOTE-061 (Shitara et al, Lancet, 2018; PMID: 29880231), 11% in ATTRACTION-2 (Kang et al, Lancet, 2017; PMID: 28993052). Thus, the molecular subtyping that we report in this manuscript identifies a cohort of patients who are much more likely to benefit from immune checkpoint inhibitors than the tools that are currently available.

In our discussion, we acknowledge that this finding, while promising, requires further validation, ideally in a prospective manner. However, we believe that this is an exciting finding that would be of great interest to the readers.

3. Patients in Group 1 had the best prognosis, could not benefit from adjuvant chemotherapy of 5-fluorouracil and platinum and have higher response rate to immune checkpoint Inhibitors. It seems that patients in Group 1 share similar features to MSI-H patients. The author should explain the difference between group 1 and MSI-H.

The reviewer raises an excellent point that Group 1 patients have similar characteristics as MSI-H patients in that both cohorts do not respond to chemotherapy but do respond to immune checkpoint inhibitors. However, only 28% of the samples in Group 1 were MSI-H, while 72% were microsatellite stable (MSS). This suggests that the MSS Group 1 patients share essential features with MSI-H patients that were identified via the 32-gene analysis. Thus, this finding emphasizes the fact that the 32-gene signature provides additional predictive power that is independent from being MSI-H.

4. The survival difference between patients in group 3 and 4 is not significant. The survival curve overlapped in the first 30 months for patients in group 2, 3 and 4.

We agree with the reviewer that in the Yonsei cohort (Fig. 2B), the most obvious survival difference was demonstrated by the improved prognosis observed in Group 1. We also agree that a simple 4-group molecular classification may be challenging to use clinically. This motivated us to build a risk score calculator that can provide a more quantitative risk score to guide care. Importantly, we were able to validate the prognostic capability of the risk score using 3 independent data sets (ACRG, Sohn et al, and TCGA). The data from the risk score is shown in Figure 3 and Supplemental Figure S3. We believe that this data demonstrates the prognostic capability, and the clinically utility, of the 32-gene signature.

5. The authors should explain the relationship between the clinical features and the classification of the gene signature. The percentage of stage IV patients was lowest in group 1 which may explain the best prognosis of group 1.

The reviewer raises an excellent point that perhaps the molecular subtyping is skewed based on stage. In Group 1, there were only 2 stage IV patients out of 114 patients, and in Group 2, there was only 3 stage IV out of 129 patients, which is a negligible difference. Groups 3 and 4 had only 7 and 8 stage IV patients, respectively, out of 162 in both groups. These small differences are unlikely to translate to the large difference in survival observed (Fig. 2B). As the Reviewer astutely noted in Comment #1, Group 1 patients resemble MSI-H patients, who are known to have improved outcomes, relative to MSS patients.

Minor points

There is no figure legend for figure 5.

We apologize for the error. The updated figure (Figure 4 in the revised manuscript) includes a legend.

Reviewer #2 (Remarks to the Author): Expert in gastric cancer and prognosis

By applying NTriPath algorithm, investigators identified signaling pathways specific to gastric cancer and 32 genes associated with top 3 signaling pathways. Authors further identified four subtype by using unsupervised clustering approach with expression data of 32 genes. Four subtypes are not only associated with overall survival of patients but also shows good correlation with benefit of standard treatments such as chemotherapy and immunotherapy. Overall, this is well designed and executed study for finding and validating novel prognostic sub types. However, there are some issues that need to be resolved before publication.

Major issues

1. Selection process for 32 genes is not clearly described. What were criteria for selecting 32 genes and cutoff for selection?

We apologize for the confusion. To clarify, we used NTriPath to analyze somatic mutations in cancer patients from 19 different types of tumors. Using NTriPath, we identified altered gene pathways that are gastric cancer-specific. In our initial report describing NTriPath (Park et al, Bioinformatics, 2016; PMID 26635139) we found that the top 3 pathways provided the most prognostic utility.

In the case of gastric cancer, the top 3 pathways are comprised of 32 genes (Supplementary Figure S1 and Supplementary Table S3). Thus, the choice of 32 genes was unbiased, and based on the genes constituting the top 3 gastric-cancer specific pathways. We have made changes to the Introduction and Methods sections of the manuscript to clarify this point.

2. Authors developed two stratification methods, 4 subtypes by clustering and 3 subtypes by generating risk score. It is not clear why they need to generate two different stratifications.

We apologize for the confusion. The 4 molecular subtypes were generated by unbiased clustering of the expression of 32 genes.

For the risk score, we arbitrarily decided to generate a low, intermediate, and high-risk group to graphically illustrate the utility of the risk score to predict survival. This is based on previous models of cancer risk estimation tools, such as Oncotype Dx for breast cancer. We now recognize rather than helping demonstrate this point, it actually confuses the issue. Thus, we have decided to remove Figure 3B.

3. Authors used clustering for validation of subtypes in independent cohorts. Clustering is not ideal approach for validation. More reliable algorithm like pamr or similar one need to be used.

We thank the reviewer for this constructive suggestion. After establishing that the 32 gene signature was prognostic for survival in the Yonsei training cohort (Figure 2B) we sought to confirm that this methodology could be applied to other datasets. As such, we analyzed other publicly available datasets to

show that subtypes obtained via unsupervised learning (i.e., consensus clustering) using the 32 genes have significantly different survival outcomes, which was indeed the case (original Figures 2C and 2D). However, it is important to note that we did not match the clusters in the different cohorts. The same color groups from different cohorts do not necessarily have similar expression patterns. We thus concluded that unsupervised learning (i.e., consensus clustering) based on the 32 gene signature provides prognostic information.

However, based on the comments from multiple reviewers, we realized that this is a confusing point. Thus, we decided to remove the consensus clustering results using ACRG and Sohn et al (original Figures 2C and 2D, and their associated Supplementary Tables). Instead, we emphasized the risk score applied to external datasets as the validation tool. We built the risk score using the Yonsei cohort and then applied the risk score to the ACRG, Sohn et al, and TCGA datasets and confirmed that the risk score robustly predicted survival. This data was shown in the original Figure 3A. For the resubmission, we moved the risk score panel as Figure 2C. Therefore, Figure 2 is entirely focused on the prognostic capability of the 32 gene signature.

For the reviewer's information, we did try to use pamr (Prediction Analysis for Microarrays) as the reviewer suggested. However, since the model does not have trainable parameters, like a weight vector in a linear regression, its prediction accuracy is low (even the training accuracy was around 65%). Thus, we chose not to include it in the manuscript.

4. Concordance with previously identified subtypes are not addressed. Previous studies including one done by authors identified 4 or 5 subtypes. Similarity or dissimilarity of current subtypes with previous ones should be directly compared as authors used same data sets.

We thank the reviewer for this constructive suggestion.

We also compared our 32-gene signature with the one that we previously published (Cheong et al, Lancet Oncology, 2018; PMID 29567071). As the Reviewer noted, both the previous study and the current study analyzed samples from Yonsei (86% overlap between the current and previous study). The Lancet Oncology study also analyzed 4 other datasets. Non-negative matrix factorization (NMF), was used to analyze transcriptomic data of the 5 datasets and 5 molecular subtypes were identified: 1) gastric subtype with high expression of gastric mucosa-specific genes (GST), 2) inflammatory subtype with immune characteristic (INF), 3) intestinal subtype with high expression of intestinal epithelial differentiation and proliferative genes (INT), 4) mixed-stromal subtype with heterogeneous transit-amplifying feature (MXD), and 5) mesenchymal subtype with mesenchymal hallmarks and stem like signatures (MSC). INF had the best prognosis, MSC had the worst, and the other 3 were intermediate and overlapped with each other.

When we compared the 4 groups identified via the 32-gene signature in this study to the 5 molecular subtypes from the Lancet Oncology study, we found that each 32-gene subgroup was represented in each Lancet Oncology subgroup. One exception was that there was a higher proportion of MSC (worst prognosis) in Group 4 (also worst prognosis). However, even in this situation, only 45.7% of MSC were in Group 4. This data is shown below and has been included as the new Supplementary Table S7.

Characteristics	Group 1 (Black)	Group 2 (Green)	Group 3 (Blue)	Group 4 (Red)	Total	P value
N	114 (20.1 %)	129 (22.8 %)	162 (28.6 %)	162 (28.6 %)	567	
Subtype in Lancet Oncology 2018						<0.001
GST	15 (13.2 %)	31 (24.0 %)	21 (13.0 %)	23 (14.2 %)	90 (15.9 %)	
INF	27 (23.7 %)	14 (10.9 %)	31 (19.1 %)	22 (13.6 %)	94 (16.6 %)	
INT	32 (28.1 %)	33 (25.6 %)	27 (16.7 %)	4 (2.5 %)	96 (16.9 %)	
MXD	30 (26.3 %)	40 (31.0 %)	14 (8.6 %)	14 (8.6 %)	98 (17.3 %)	
MSC	2 (1.8 %)	2 (1.6 %)	32 (19.8 %)	74 (45.7 %)	110 (19.4 %)	
Missing	8 (7.0 %)	9 (7.0 %)	37 (22.8 %)	25 (15.4 %)	79 (13.9 %)	

Finally, when we included the Lancet Oncology subtype as part of a multivariate analysis, we found that the 32-gene signature continues to be independently associated with overall survival. This data is shown below and has been included as the new Supplementary Table S8.

Characteristics	Hazard Ratio (95% CI)	P value
Age		
≤60 years	Reference	
>60 years	2.01 (1.52, 2.65)	<0.001
Stage		
I	Reference	
II	1.44 (0.50, 4.11)	0.5
III	3.25 (1.19, 8.89)	0.022
IV	16.94 (5.54, 51.76)	<0.001
Lauren Type		
Diffuse	Reference	
Intestinal	0.94 (0.66, 1.33)	0.728
Mixed	0.71 (0.33, 1.56)	0.4
Other	1.32 (0.92, 1.90)	0.128
Perineural Invasion		
Negative	Reference	
Positive	1.15 (0.81, 1.62)	0.442
Subtype in Lancet Oncology 2018		
GST	Reference	
INF	0.71 (0.44, 1.14)	0.158
INT	0.96 (0.61, 1.52)	0.857
MXD	1.47 (0.95, 2.28)	0.085
MSC	1.03 (0.66, 1.59)	0.912
Molecular Subtype		
Group 1	Reference	
Group 2	1.96 (1.25, 3.07)	0.003
Group 3	1.64 (1.05, 2.57)	0.03
Group 4	2.41 (1.48, 3.94)	<0.001

We also compared the 32-gene signature to the classification system published by the ACRG (Cristescu, Nature Medicine, 2015; PMID 25894828). We found that Group 4 (worse survival) was particularly enriched in the MSS/EMT group (worse survival). Group 1 (best survival) was also enriched for MSI (best survival) but 45% of MSI samples were found in Groups 2-4 so it was not a direct correlation. Finally, the MSS/TP53+ and MSS/TP53- groups had more even distributions across Groups 1-4. This data is shown below and has been included in the new Supplementary Table S7

Characteristics	Group 1 (Black)	Group 2 (Green)	Group 3 (Blue)	Group 4 (Red)	Total	P value
ACRG						
MSS/EMT	0 (0.0 %)	6 (8.5 %)	0 (0.0 %)	40 (62.5 %)	46 (15.3 %)	<0.001
MSI	43 (55.1 %)	10 (14.1 %)	12 (13.8 %)	3 (4.7 %)	68 (22.7 %)	
MSS/TP53+	14 (17.9 %)	23 (32.4 %)	33 (37.9 %)	9 (14.1 %)	79 (26.3 %)	
MSS/TP53-	21 (26.9 %)	32 (45.1 %)	42 (48.3 %)	12 (18.8 %)	107 (35.7 %)	

Thus, the 32-gene signature was not a simple recapitulation of the previously published classification systems and provides new information. In addition, subtyping based on the 32-gene signature provides predictive information regarding response to immune checkpoint inhibitors, which is completely novel. We discussed this data in the Results section under the “Identification of a Prognostic 32-Gene Signature and Molecular Subtypes” subsection.

Reviewer #3 (Remarks to the Author): Expert in machine learning

The authors propose and study a 32-gene signature for prognosis and prediction of response to therapy in gastric cancer. The approach builds on previous work (Park et al., Bioinformatics 2016; ref #11 in the current ms) that put forward a tool (NTriPath) aimed at identifying pathways associated with survival. The main novelty of the current manuscript lies in the application to gastric cancer.

The general workflow is as follows. Using large pan-cancer data (from TCGA) the previously established NTriPath is used to identify a gastric cancer-specific signature whose association with prognosis and therapeutic outcome is studied using clinical data.

The paper is well motivated and addresses an interesting question. It is nice to see work with a specific clinical focus and the authors have made a good effort to bring together several clinical datasets.

However, a number of questions concerning the data analysis should be addressed and clarified, as detailed below, and potential limitations stated clearly. This is particularly important in this domain, as care is needed in the use and interpretation of machine learning methods to inform clinical practice.

Main points:

1. Clarify dataset use for analysis steps. The manuscript as written is not fully clear with respect to how various datasets were used for different steps in the analysis. Please see detailed comments below for specific examples with line-by-line references, but in general it would be important throughout the paper to make this information absolutely as clear as possible. Each of the steps (gene selection, clustering, learning prediction models etc.) involves learning from data in some form and is important to be clear about what parameters were trained on which data and which parameters were then carried across to other datasets.

We thank the reviewer for this suggestion. We edited Figure 1 to better delineate the workflow of the study.

2. Selection bias. The results concerning prediction of response to therapy are based on non-randomized data (lines 212-216). This may result in bias relevant to the results presented. The authors note this point in Discussion (line 312), but potential bias and its importance for clinical interpretation should be made clear throughout. You note that patients were included who were treated prior to adjuvant therapy becoming standard of care. If known, can you say a little more about how treatment decisions were made, i.e. based on which factors? This may help readers to think about potential selection bias.

As the reviewer correctly points out, one of the limitations of retrospective analyses is selection bias. The choice to undergo adjuvant therapy before it had become standard of care was likely driven by a combination of physician and patient factors. These may include biases against chemotherapy (e.g., some patients do not want to receive chemotherapy because their friends had bad experiences with it), in favor of chemotherapy (e.g., patients who want “everything possible to be done” to treat their cancers), and patient status (e.g., a patient is considered to be too old or too weak to receive chemotherapy), etc.

We included a discussion of this bias in the Discussion and the need for validation of our findings in a prospective and randomized manner to mitigate these limitations:

“However, our current study is limited by the retrospective nature of our analysis that may be confounded by selection bias. For example, our training set was based on samples obtained before adjuvant chemotherapy was standard of care. Thus, the decision to refer a patient for post-operative chemotherapy versus observation are unknown. Similarly, while we showed that the molecular classifier was associated with increased response to immune checkpoint inhibitors, response may not accurately predict overall survival. Thus, the prognostic and predictive utility of the molecular classification scheme should be validated prospectively using the most clinically relevant endpoints, such as overall survival and/or quality of life.”

3. Clarify small-sample issues. In places, the models appear to be used in very small sample settings. For example, in the immune checkpoint analysis (line 230 onwards), n=45 patients were available. From the current description, it appears that a multi-class SVM was trained using these data. If so, this is a very small number with which to train and test a classifier. Furthermore, this includes the four subtypes, hence the subtype-specific sample sizes must be even smaller. Please clarify this point and report numbers of patients in the main text itself and not just percentages.

We apologize for the confusion, as our original text did not clearly describe our methodology. To clarify, the training did not occur with the 45-patient group published by Kim et al. Rather, we established the 4 subgroups using the 567-patient cohort from Yonsei. Then, each patient treated with immune checkpoint inhibitors was classified into one of the four subgroups and the proportion of responders and non-responders between groups was compared. For the resubmission, we added 17 additional patients, so now there are 90 total patients. For the resubmission, we also chose to pool all of the patients into one analytic cohort, and this is now presented in the new Figure 4 (see below). The number of patients has been included in the text.

Figure 4

Detailed comments:

4. Abstract. This should be edited in light of points 1-3. In particular please note the potential for selection bias and clarify the role of initial and validation datasets.

Thank you for the suggestions. In the abstract, we did note that this study was retrospective in nature, which connotes the potential for selection bias. However, due to the strict word limit there is no space for a more detailed explanation, which we did provide in the Discussion section. We did add wording to clarify the roles of the various datasets, as suggested by the Reviewer

5. Line 143: please clarify precisely which expression data were used for the consensus clustering.

We have clarified this line:

“We performed consensus clustering of the 567-patient Yonsei cohort based on the expression level of the 32 genes...”

6. Line 164. The multivariable Cox analysis uses "variables that were significant on univariable analysis". What data were the univariate and multivariable models respectively fit on? If the same dataset for both, then are the latter estimates biased (due to the selection step)?

We thank the reviewer for pointing out this. Indeed, both univariate and multivariate analysis models were fit on the Yonsei cohort. As the reviewer pointed out, the results from the multivariate analysis could be biased due to the selection step using the univariate analysis. To address this possible selection bias, we conducted a regularized Cox regression (Cox model with the lasso regularization, implemented with the glmnet package in R).

Using the Yonsei cohort, we set all clinical variables (except for the variables with a high proportion of missing data, such as EBV status, which was missing in ~60% of samples) as predictors and let the model itself select the optimal features for survival. We also calculated the 95% confidence interval of the hazard ratio and the p-value of each variable using bootstrapping (the null hypothesis H_0 is that each coefficient $\beta_j = 0$). We ran the model 10,000 times in the bootstrapping setting (the training data in each run was resampled with replacement) and calculated the hazard ratio interval and the p-value of each coefficient based on the empirical distribution of the estimator of the coefficient. Only three variables p-values < 0.05 were selected: age, stage, and our molecular subtypes. This result is similar to what we obtained from the multivariate analysis.

We added additional text to the Results and Methods to address these points. The results from the regularized Cox regression are in the new supplementary tables S5 and S6.

7. Line 170 onwards, subsection "Validation of the 32-gene Prognostic Signature to Discover molecular subtypes". It is stated: "Unsupervised consensus clustering using the 32-gene signature again identified four molecular subtypes". Was the clustering re-done using the ACRG and Sohn et al. datasets in the sense that consensus clustering was performed using the 32 genes previously identified, but using expression data from these two studies? If so, it should be noted that a new clustering using the same genes will not in general give the same partition of the feature space, i.e. the clusters in the validation data may occupy different regions of gene expression space than in the original, discovery dataset. Furthermore, can you explain how the clusters were matched up, i.e. how did you match a specific cluster discovered in ACRG, say, with a cluster in the original data?

We apologize that our methods description was not clear in the original submission. In fact, Reviewer 2 asked a similar question.

After establishing that the 32 gene signature was prognostic for survival in the Yonsei training cohort (Figure 2B) we sought to confirm that this methodology can be applied to other datasets. We analyzed other publicly available datasets to show that subtypes obtained via unsupervised learning (i.e., consensus clustering) using the 32 genes have significantly different survival outcomes, which was indeed the case (original Figures 2C and 2D). However, it is important to note that we did not match the clusters in the different cohorts. The same color groups from different cohorts do not necessarily have similar expression patterns. We thus concluded that unsupervised learning (i.e., consensus clustering) based on the 32 gene signature provides prognostic information.

However, based on the comments from multiple reviewers, we realized that this is a confusing point. Thus, we decided to remove the consensus clustering results using ACRG and Sohn et al (original Figures 2C and 2D, and their associated Supplementary Tables). Instead, we used the risk score applied to external datasets as the validation tool. We built the risk score using the Yonsei cohort then applied the risk score to the ACRG, Sohn et al, and TCGA datasets and confirmed that the risk score robustly predicted survival. This data was shown in the original Figure 3A. For the resubmission, we moved the risk score panel as Figure 2C so now, Figure 2 is entirely focused on the prognostic capability of the 32 gene signature.

8. SVM learning, Line 193. It is stated that a SVM was trained "to estimate five-year overall survival." However in line 195 it is stated that a binary classifier was learned, using labels for groups 1,4 as training labels. Is the idea to use these labels to focus on extremes, rather than directly modelling survival?

We apologize for the confusion. We did not directly model survival. The binary classifier was trained using only Groups 1 and 4 and their group cluster labels in the Yonsei cohort. As mentioned by the reviewer, Groups 1 and 4 were interpreted as extremes (Group 4 was treated as the positive class). We have updated the main text to make this point clearer. The detailed descriptions of how to build the

prediction model (a linear SVM) for risk score is in the Methods section under “Predictive model for risk scores to predict overall survival”.

Later on it is stated that using the validation data (ACRG and Sohn et al.) "the risk score, as a continuous variable, was prognostic of five-year overall survival". It is not clear what is meant here: can you rephrase?

We apologize for the confusion; we have edited the text. Essentially, we meant to say the risk score is prognostic for survival.

9. Added value compared with clinical features, line 205. In the validation data, can you compare the fitted SVM with a regression using only the clinical and pathological features? This would help to directly quantify added value in terms of held-out classification/prediction accuracy.

We thank the reviewer’s suggestion. However, unfortunately, there were not many clinical or pathological variables available to build a regression model for survival prediction in the validation data. Most of the clinical and pathological features have a high proportion of missing values or do not have the exact same format across the datasets making the suggested comparisons difficult. The only available variables for this purpose in the datasets are age, sex, and stage.

We trained a logistic regression using these three variables in the Yonsei Cohort (Group 1 is treated as the negative class and Group 4 as the positive class) and tested it on the ACRG and Sohn et al datasets. Here we used a logistic regression as a predictor, instead of an SVM, because most variables are categorical. As shown in the figures below, we divided the samples into the three risk groups based on the risk scores in each dataset (low risk was defined as a risk score of less than the 25th percentile, intermediate risk as a score of equal or higher than the 25th percentile but less than the 75th percentile and high risk as a score of equal or higher than the 75th percentile). For both datasets, there is no statistically significant separation between the three groups.

Thus, based on this data alone, one would conclude that the clinical and pathological features do not provide more information for survival prediction, which we know to be false. Thus, we attribute the finding to the amount of missing information in the validation datasets that prevents us from performing a robust analysis. Thus, we chose not to include this analysis in the manuscript.

Kaplan-Meier curves for overall survival stratified by risk group in the ACRG cohort

Kaplan-Meier curves for overall survival stratified by risk group in the Sohn et al cohort

10. Line 212. It would be relevant to point out the possibility of selection bias here (see #2 above).

We have included a more detailed discussion of the possibility of selection bias in the Discussion.

11. Line 224. Please state absolute numbers of patients and not only HR.

We have made this change.

12. Line 230. Using the Kim et al. data how were the patients clustered into groups 1-4 and how were the clusters matched with those from Yonsei? It is stated "There were 45 patients with RNA-seq data available for analysis. Using that data, we built a multi-class SVM..." Does this mean an SVM was trained on n=45 data covering 4 classes? How was this done? A sample size of n=45 is very small to train and test a classifier.

We apologize for the confusion, as our original text did not clearly describe our methodology. To clarify, the training did not occur with the 45-patient group published by Kim et al. Rather, we established the 4 subgroups using the 567-patient cohort from Yonsei. Then, each patient treated with immune checkpoint inhibitors was classified into one of the four subgroups based on their expression level of the 32 genes and then we compared the proportion of responders and non-responders. For the resubmission, we added 17 more patients, so now there are 90 total patients. For the resubmission, we also chose to pool all the patients into one analytic cohort, and this is now presented in the new Figure 4 (see below).

14. Line 237. Please state patient numbers and not only proportions. With n=45 over four groups, how many patients were in each group?

After analyzing 17 more patients treated with immunotherapy, we now have a total of 90 patients. Group 1 had an overall response rate (ORR) to immune checkpoint inhibitors of 48% (N=21) and Group 3 had an ORR of 50% (N=14). An ORR was observed in only 8% of Group 2 patients (N=24) and 13% of Group 4 patients (N=31). This difference was statistically significant ($p = 0.001$).

In the original report, we presented the two groups as separate cohorts. As part of a response to comment 2 by Reviewer 1, we decided to pool all of the cohorts into one analysis for the resubmission. Overall, we believe the data is convincing for the molecular subtyping that we identified to be associated with response to immune checkpoint inhibitors. Both patient numbers and proportions are now stated in the text.

15. Line 243. Please address the same questions in #14,15 also for the n=28 Seoul St Mary's Hospital data.

Please see the response to comment 14.

16. Discussion. Can you state how you would expect the models to be used in practice? What would be the clinical workflow? What next steps would be needed to take these ideas forward? This would be very useful to contextualise the work and give readers a clear sense of the scope.

Thank you for this suggestion. We believe that our model has the potential to improve the precision of gastric cancer care. In addition to providing patients and providers with prognostic information, the 32 gene signature also has utility as a predictive biomarker to help guide therapy. In the adjuvant setting, the 32 gene signature identifies patients most likely to benefit from chemotherapy. Additionally, our model can assist clinicians with selection of patients who are likely to derive benefit from immune checkpoint blockade. To reach these clinical applications, we would develop a 32 gene assay based on qRT-PCR or nCounter analysis system. We have included a more robust contextualization of our findings throughout the manuscript.

17. Gene-set analysis, line 341. Please state what statistical test was used within gene-set analysis.

We used the PANTHER (Protein ANalysis THrough Evolutionary Relationships) Classification System using Fisher Exact Test with FDR < 0.05 (<http://www.pantherdb.org>) for gene set enrichment (Mi et al, Nucleic Acids Res, 2020; PMID 33290554). This information and reference were added to the Methods.

REVIEWERS' COMMENTS

Reviewer #1 (Remarks to the Author):

The authors try to identify a gene signature which may be independently associated with overall survival and predict response to chemotherapy and PD-1 inhibitors. However, the number of patients who were in stage IV was small. There is no need to predict response to chemotherapy and PD-1 inhibitors in stage I-III patients. Moreover the number of patients who received immunotherapy was too small. It is not sufficient to predict the response to PD-1 inhibitors.

The authors mentioned that this signature can identify patients who may have a 50% ORR to PD-1 inhibitors. Actually, without this signature, only use MSI-H single marker, this group of patients can also get a 40-50% ORR to PD-1 inhibitors.

Reviewer #2 (Remarks to the Author):

no further concerns

Reviewer #3 (Remarks to the Author):

Thank you very much for the revision and detailed point-by-point response. The manuscript is, in my view, improved as a result of the changes made, with a sharper focus and clearer discussion of the limitations.

Some further points are noted below:

* Need for further prospective evaluation. While this is now noted in the abstract (line 82), there are other places in the text where readers might miss this important point. For example, line 256 could be rephrased from "powerful new risk stratification tool to optimize treatment choice..." to "powerful new risk stratification tool with the potential to optimize treatment choice..." with a note on the the need for further prospective evaluation.

* Line 156: here the bias is with respect to variable selection. To avoid confusion with selection bias in the context of response to therapy, it may be better to replace "selection bias" with simply "bias" here.

Reviewer #1 (Remarks to the Author):

The authors try to identify a gene signature which may be independently associated with overall survival and predict response to chemotherapy and PD-1 inhibitors. However, the number of patients who were in stage IV was small. There is no need to predict response to chemotherapy and PD-1 inhibitors in stage I-III patients.

To test the ability of the 32 gene signature to predict response to immune checkpoint inhibitors, we analyze 90 patients with recurrent or metastatic gastric cancer. No stage I-III patients were included in the cohort (other than the ones who developed recurrence).

While immune checkpoint inhibitors were initially approved for patients with advanced disease, there is an increasing amount of evidence to support its use in earlier stage disease. For example, the recently published CheckMate 577 phase 3 randomized control trial (Kelly et al, NEJM, 2021; PMID: 33789008) demonstrated that nivolumab improved survival in patients with stage II or III gastroesophageal junction (along with esophageal) cancers in the adjuvant setting.

Moreover the number of patients who received immunotherapy was too small. It is not sufficient to predict the response to PD-1 inhibitors. The authors mentioned that this signature can identify patients who may have a 50% ORR to PD-1 inhibitors. Actually, without this signature, only use MSI-H single marker, this group of patients can also get a 40-50% ORR to PD-1 inhibitors.

We agree with the Reviewer that MSI-H does predict response to immune checkpoint inhibitors. However, importantly, a significant number of Group 1 and 3 patients were *NOT* MSI-H, suggesting that the 32-gene signature provides predictive capability for non MSI-H patients, independent of MSI status.

We believe that our findings are an important "signal" that justifies further study involving a larger patient cohort.

Reviewer #2 (Remarks to the Author):
no further concerns

Reviewer #3 (Remarks to the Author):

Thank you very much for the revision and detailed point-by-point response. The manuscript is, in my view, improved as a result of the changes made, with a sharper focus and clearer discussion of the limitations.

Some further points are noted below:

*** Need for further prospective evaluation. While this is now noted in the abstract (line 82), there are other places in the text where readers might miss this important point. For example, line 256 could be rephrased from "powerful new risk stratification tool to optimize treatment choice..." to "powerful new risk stratification tool with the potential to optimize treatment choice..." with a note on the the need for further prospective evaluation.**

Thank you for this suggestion, we have improved this area of the discussion accordingly.

*** Line 156: here the bias is with respect to variable selection. To avoid confusion with selection bias in the context of response to therapy, it may be better to replace "selection bias" with simply "bias" here.**

Thank you for this comment, we agree and have improved the text as suggested.